# Exosomal miR92a Promotes Cytarabine Resistance in Myelodysplastic Syndromes by Activating Wnt/β-catenin Signal Pathway

**DOI:** 10.3390/biom12101448

**Published:** 2022-10-09

**Authors:** Hongjiao Li, Chenglian Xie, Yurong Lu, Kaijing Chang, Feng Guan, Xiang Li

**Affiliations:** 1Key Laboratory of Resource Biology and Biotechnology of Western China, Ministry of Education, Provincial Key Laboratory of Biotechnology, College of Life Sciences, Northwest University, Xi’an 710069, China; 2Institute of Hematology, School of Medicine, Northwest University, Xi’an 710069, China

**Keywords:** MDS/AML, exosomal miR92a, Ara-C resistance, PTEN, Wnt/β-catenin

## Abstract

Cytarabine (Ara-C) has been one of the frontline therapies for clonal hematopoietic stem cell disorders, such as myelodysplastic syndrome (MDS) and acute myeloid leukemia (AML), but Ara-C resistance often occurs and leads to treatment failure. Exosomal microRNAs (miRNAs, miRs) as small noncoding RNA that play important roles in post-transcriptional gene regulation, can be delivered into recipient cells by exosomes and regulate target genes’ expression. miR92a has been reported to be dysregulated in many cancers, including MDS and AML. However, the effects of exosomal miR92a in hematologic malignancies have not been fully investigated. In this study, qualitative analysis showed the significantly enhanced expression of exosomal miR92a in MDS/AML plasma. Subsequent functional assays indicated that exosomal miR92a can be transported and downregulate PTEN in recipient cells and, furthermore, activate the Wnt/β-catenin signaling pathway and interfere with the Ara-C resistance of receipt MDS/AML cells in vitro and in vivo. Altogether, our findings offer novel insights into plasma exosomal miR92a participating in Ara-C resistance in MDS/AML and we propose miR92a as a potential therapeutic target for MDS/AML.

## 1. Introduction

Myelodysplastic syndromes (MDS) comprise a heterogeneous group of myeloid neoplasms, which are characterized by peripheral blood cytopenia, hematopoietic cell dysplasia, and a variable disease course. About 30% of MDS patients are at high risk of transformation to secondary acute myeloid leukemia (AML) [1]. As one of the common chemotherapy drugs used for MDS and AML therapy, cytarabine (Ara-C) is used clinically in combination with anthracyclines to achieve the best therapeutic effect [2,3,4]. Ara-C enters the cell and is converted into a therapeutically active triphosphate metabolite, Ara-CTP, which enters the nucleus and inhibits DNA synthesis, in turn triggering apoptosis and exerting antileukemic effects [5]. However, most of the MDS/AML patients will develop Ara-C resistance, and not respond to subsequent therapy. Therefore, understanding the molecular mechanisms that contribute to the emergence of Ara-C resistance is required for improving therapeutic outcomes.

Accumulating evidence supports the idea that the continuous cross-talk between cancer cell and local/distant environments is required for effective tumor growth and systemic dissemination, having an important role in the development of drug resistance [6,7]. Exosomes are nano-sized membrane-covered structures (with a diameter of 30–150 nm), originated from the endosomal pathway and secreted via exocytosis into extracellular space [8,9]. Exosomes can mediate intercellular communication between donor and recipient cells by enriching and delivering nucleic acids and proteins [10]. Especially, exosomes act as vehicles for exchange of microRNAs (miRNAs) between heterogeneous populations of tumor cells, generating a transmitted drug resistance [11,12]. miRNAs are a class of small noncoding RNAs, which regulate gene expression by binding to the 3′ untranslated regions (UTR) of target mRNAs [13]. Exosomes can enwrap miRNAs, protect them from degradation, deliver them to recipient cells and modulate tumor immunity and the surrounding microenvironment, further facilitating tumor growth, invasion, metastasis, angiogenesis and drug resistance [14,15,16].

miR92a, located on human chromosome 13q32-33, has been studied in gastric cancer, lung cancer, prostate cancer, liver cancer and thyroid cancer [17,18,19,20]. For example, it suppressed Dickkopf 3 (DKK3) transcription to enhance migration and invasion in colon cancer [21,22]. It also promoted cell proliferation by targeting tumor-suppressor gene F-box and WD-40 domain protein 7 (FBXW7) in nonsmall cell lung cancer [21,22]. These data implied the carcinogenesic role of miR92a. Circulating miR92a in plasma has been thought as novel potential biomarkers for AML [23,24]. However, the function of exosomal miR92a in MDS and AML clone cells was still unclear. In this study, we found that the expression of exosomal miR92a in MDS/AML plasma were higher than that in healthy donor (HD). When recipient cells were treated with exosomes derived from MDS/AML plasma, we found exosomes that expressed high miR92a induced a stronger proliferation ability and Ara-C tolerance of recipient cells.

## 2. Results

### 2.1. Upregulated Exosomal miR92a in MDS/AML Plasma

Firstly, we analyzed the prognosis survival curve of miR92a in pan cancer via the TCGA database. The results showed that higher expressions of miR92a-1 and miR92a-2, two precursors of miR92a, resulted in a poor survival rate depending on Kaplan–Meier analysis (Figure 1A). In addition, miR92a expression in primary blood-derived cancer was higher than that of other sample types (Figure 1B).

Next, we evaluated the expression of exosomal miR92a in MDS/AML plasma. The exosomes (Exos) from plasma of healthy donors (HD) and MDS/AML patients were isolated via differential centrifugation and ultracentrifugation (Figure 1C). The purified Exos presented a stable expression of marker proteins (Alix, CD81, CD63 and TGS101) by Western blotting (Figure 1D), and the typical characteristic shapes and sizes by transmission electron microscopy (TEM) and nanosight tracking analysis (NTA) (Figure 1E,F). Next, the qRT–PCR results showed that the expression of exosomal miR92a in MDS/AML plasma was significantly higher than that in HD (Figure 1G). According to the exosomal miR92a level in patient plasma compared to that in HD plasma, Exos derived from MDS/AML patient plasma were divided into two groups, high miR92a-Exos and low miR92a-Exos (Appendix A). When SKM1 cells were treated with these two groups of Exos, the miR92a level was significantly upregulated in recipient SKM1 cells (Figure 1H). Meanwhile, high miR92a-Exos and low miR92a-Exos induced resistance to Ara-C in recipient SKM1 cells, and low miR92a-Exos induced SKM1 Ara-C resistance to a relatively lower extent than high miR92a-Exos (Figure 1I). Above all, we speculated that exosomal miR92a is correlated to MDS progression and resistance to Ara-C.

### 2.2. Effect of Exosomal miR92a on Recipient Cell Ara-C Resistance

To explore the function of exosomal miR92a, we overexpressed hsa-miR-92a-1 and hsa-miR92a-2 in SKM1, termed as SKM1-miR92a1 and SKM1-miR92a2 (Figure 2A,B). When SKM1-miR92a1/2 cells were treated with 2 μM Ara-C, apoptosis rates were lower than parental SKM1 cells (Appendix A).

Next, exosomes from SKM1, SKM1-miR92a1/2 (termed as Exos-SKM1, Exos-miR92a1 and Exos-miR92a2) were isolated by differential centrifugation and characterized by Western blotting, TEM and NTA (Appendix A). Flow cytometric results indicated that Exos-SKM1, Exos-miR92a1 and Exos-miR92a2 can be uptaken by recipient SKM1 (Figure 2C). Meanwhile, compared to Exos-SKM1, Exos-miR92a1 and Exos-miR92a2 treatment increased the miR92a level in recipient SKM1 (Figure 2D). Under Ara-C treatment, decreased apoptosis in SKM1 was found after Exos-miR92a1/2 treatment (Figure 2E). Meanwhile, Exos-miR92a1/2-treated SKM1 presented increased proliferation with Ara-C treatment (Figure 2F). Using another MDS cell line (ML1) as recipient cells, similar results of cell apoptosis and proliferation with Ara-C treatment were observed (Appendix A).

### 2.3. miR92a Binds to Target Gene PTEN

To determine the mechanism of miR92a in mediating Ara-C resistance, we browsed biological target genes of miR-92a using three online servers (TargetScan, miRDB and TarBase) and screened out five potential target genes, ZEB, FGF2, SMAD4, KLF4 and PTEN (Figure 3A). qRT-PCR analysis showed the decreased expressions of ZEB, FGF2, SMAD4, KLF4 and PTEN in SKM1-miR92a1 and SKM1-miR92a2 cells (Figure 3B). Among these five genes, PTEN expression significantly decreased and it was reported to correlate with drug resistance [15,25]. Therefore, we aimed at PTEN as the potential target of miR92a (Figure 3B). In addition, potential binding sites of miR-92a were identified in the PTEN 3′UTR based on the databases above (Figure 3C). The potential binding capacity of miR92a to PTEN was evaluated by inserting a wild-type (WT) or mutant 3′-UTR sequence of PTEN downstream of the luciferase reporter gene in HEK-293T. The dual-luciferase reporter gene assay showed that the activity of luciferase in miR92a-mimic-treated HEK-293T cells was significantly reduced by inserting a PTEN WT sequence but unaffected by inserting a mutant 3′-UTR PTEN sequence (MUT) (Figure 3D). TCGA database analysis further confirmed a negative relation between miR-92a and PTEN expression in AML (*n* = 206) and HD (*n* = 18) (Figure 3E).

### 2.4. miR92a Regulates Ara-C Resistance via Activating Wnt/β-Catenin Signal

It has been found that PTEN is associated with the Wnt/β-catenin pathway in drug resistance [26,27]. The activated Wnt signaling pathway activates gene transcription through transporting β-catenin into the nucleus [28]. We then explored β-catenin activation to reveal potential mechanisms underlying the role of miR-92a in increasing Ara-C resistance. The results showed that the increased expression of β-catenin and target genes, Axin2, c-Myc and CyclinD1, of β-catenin was found in Exos-miR92a1/2-treated SKM1 and ML1 (Figure 4A,B and Appendix A). Furthermore, it showed decreased PTEN and increased β-catenin in the nucleus of SKM1 and ML1 after treatment with Exos-miR92a1/2 (Figure 4B and Appendix A). Next, the TCGA database indicated increased expression of β-catenin in AML samples (*n* = 173) compared to HD (*n* = 70) (Figure 4C). Interestingly, when SKM1 and ML1 were incubated with a selective inhibitor of β-catenin, MSAB [15], the apoptosis of Exos-miR92a1/2-treated recipient cells was increased significantly compared to Exos-SKM1, which revealed that MSAB treatment reversed the Ara-C resistance of SKM1 and ML1 caused by Exos-miR92a1/2 (Figure 4D and Appendix A). Together, these data indicate that exosomal miR92a mediates the Ara-C resistance of SKM1 via inhibiting PTEN and activating β-catenin.

### 2.5. Exosomal miR92a Promoted MDS/AML Ara-C Resistant In Vivo

To detect the effect of exosomal miR92a on Ara-C resistance in vivo, we injected mice with SKM1 cells in combination with Ara-C and Exos-miR92a1/2 (Figure 5A). Flow cytometry results showed that the injection of Exos-miR92a1/2 can significantly increase SKM1 proliferation in vivo after treatment with Ara-C (Figure 5B). In addition, immunohistochemistry (IHC) staining showed increased β-catenin and decreased PTEN in the spleen and bone marrow of mice after injecting Exos-miR92a1/2 (Figure 5C).

## 3. Discussion

Ara-C has been the commonly used therapeutic agent for MDS and AML patients for decades. Considerable progress has been made in the development of new treatments for MDS/AML patients, but drug resistance remains a major clinical problem. Increasing evidence indicates the involvement of exosomes in mediating drug resistance through several mechanisms [26,29,30]. Drug-resistant cancer cells are able to pack the chemotherapeutic agents in exosomes and shuttle anti-cancer drugs out of tumor cells [31], or deliver exosomal cargoes containing miRNA and proteins that later induce drug resistance to recipient cells [32]. For example, exosomal miR21 enhanced cisplatin resistance by suppressing the inflammasome activity of NOD-like receptor thermal protein domain-associated protein 3 (NLRP3) [33]. Exosomal miR155 induces chemoresistance through increasing epithelial–mesenchymal transition (EMT) markers in breast cancer [34].

Previous studies have shown that miR92a plays a crucial role in the development of multiple cancers and the dysregulation of miR92a is reported to have potential as a tumor biomarker [17,35]. However, the function of exosomal miR92a during hematological malignant progression remains unclear. It was reported that hepatocellular carcinoma cell (HCC)-derived exosomal miR92a can promote EMT and convert low-metastatic HCCs into high-metastatic HCCs [18]. Moreover, an elevated expression of exosomal miR92a in plasma is positively correlated with the metastasis of HCC [18]. miR92a-enriched exosomes derived from cancer-associated fibroblasts of colorectal cancer (CRC) can be transferred into CRC cells to promote migration, invasion, metastasis, stemness, and drug resistance [36]. Using the TCGA database, we found that the miR92a level was upregulated in pan cancer and an increased level of miR92a is related to poor cancer prognosis. Later, we demonstrated that the exosomal miR92a level was significantly increased in the plasma of MDS/AML patients. Exosomes are not only the cargoes that contain circulating RNAs, DNAs or proteins, but also reflect the pathological statues of originated cells or tissues. The aberrant exosomal miR92a in MDS/AML also suggests that potential donor cells, such as malignant clonal cells, endothelial cells or macrophages, which generate the dysregulated exosomes, need to be investigated in future studies.

Our data indicate both high levels of miR92a-Exos and low levels of miR92a-Exos can induce recipient cells that are resistant to Ara-C, and low levels of miR92a-Exos also induced SKM1 Ara-C resistance to a relatively lower extent than high levels of miR92a-Exos. We speculated that miR92a may be the dominant factor in high miR92a-Exos-induced Ara-C resistance of recipient cells, while some other mechanisms may exist in low miR92a-Exos-mediated Ara-C resistance.

miRNAs can post-transcriptionally suppress target mRNA expression, mostly through interaction with 3′ UTR. Using the database and molecular assay, we found that PTEN served as a target of exosomal miR92a and the resulting downregulation of PTEN promoted Ara-C resistance in recipient cells via Wnt/β-catenin pathway. PTEN is a ubiquitously expressed tumor suppressor that is commonly inactivated in human sporadic cancers and it is also major negative regulator of the AKT signaling pathway and Wnt/β-catenin signaling pathway [37,38]. Deficient PTEN suppressed by HOX transcript antisense RNA confers adriamycin resistance in AML [39]. Additionally, PTEN deficiency can activate the AKT pathway to sustain refractory AML status through the enhancement of glycolysis and mitochondrial respiration [40]. A recent study revealed the involvement of PTEN and the Wnt/β-catenin pathway in drug resistance [27], and it has been found that PTEN overexpression can block β-catenin-induced urothelial proliferation and tumorigenesis [41]. In addition, the Wnt/β-catenin pathway has been proved to be required for the development of leukemic stem cells in AML [42] and inactivated Wnt/β-catenin can suppress proliferation and P-gp-mediated multidrug resistance in AML [43]. Taken together, our data indicated that interference with the expression of PTEN and the Wnt/β-catenin signaling pathway may reverse the Ara-C resistance caused by exosomal miR92a.

In conclusion, this study revealed that plasma exosomal miR92a is upregulated in MDS/AML and can be transport to receipt cells. It can regulate the Ara-C sensitivity of recipient cells by inhibiting PTEN and activating β-catenin. These results indicate that exosomal miR-92a is a potentially useful target for a more effective chemotherapy of MDS/AML patients.

## 4. Materials and Methods

### 4.1. Patient Samples

The plasma of heathy donors and MDS/AML patients was obtained from Shanxi Provincial People’s Hospital. The patients’ information is listed in Appendix A. Written informed consent was obtained from all patients in accordance with the Declaration of Helsinki. Experiments using human tissues were approved by the Research Ethics Committee of Northwest University.

### 4.2. Exosomes Extraction

Exosomes were isolated as described previously [44]. In brief, culture supernatants were collected and subjected to successive centrifugations at 300× *g* for 10 min, 2000× *g* for 10 min, 10,000× *g* for 30 min, and 100,000× *g* for 70 min at 4 °C. Exosome pellets were rinsed with PBS, collected by ultracentrifugation at 100,000× *g* for 70 min (Optima XE-100 ultracentrifuge; Beckman Coulter Life Sciences; Indianapolis, IN, USA), resuspended in 100 μL PBS, and stored at −80 °C. Exosomes were labeled using Exo-tracker, as described previously [45]. Exosomes were stained with 10 μM Exo-tracker for 30 min at 37 °C and collected by ultracentrifugation. For the purification of exosomes from AML/MDS patient and HD plasma, plasma was subjected to successive centrifugations at 2000× *g* for 30 min, 12,000× *g* for 45 min, and 110,000× *g* for 2 h at 4 °C. Pellets were resuspended in PBS, filtered (pore size 0.22 μm), collected by ultracentrifugation at 110,000× *g* for 70 min, and resuspended in 100 μL PBS.

### 4.3. Transmission Electron Microscopy (TEM)

Purified exosomes were applied to carbon-coated 400 mesh grids (Electron Microscopy Sciences; Fort Washington, PA, USA) for 5 min, washed with PBS, and stained with 2% uranyl acetate for 30 s, as described previously [46]. Images were obtained by TEM (model H-7650; Hitachi; Tokyo, Japan) at 80 kV.

### 4.4. Nanosight Tracking Analysis (NTA)

Exosomes were loaded into a NanoSight LM10 instrument (Malvern; UK) and particles were tracked for 60 s using the NanoSight nanoparticle tracking analysis software program.

### 4.5. Cell Culture

The human hematopoietic cell lines SKM1 and ML1 were cultured in RPMI 1640 medium (Biological Industries, Beit Haemek, Israel) and kidney epithelial cell HEK-293T cells were cultured in DMEM medium (Biological Industries) supplemented with 10% Fetal Bovine Serum (FBS) (Biological Industries) and 1% penicillin/streptomycin (Beyotime Biotechnology, Haimen, Jiangsu, China) at 37 °C in a 5% CO_2_ atmosphere.

### 4.6. Stable Transfection

A human precursor of miR92a1 and miR92a2 was cloned from SKM1 cells. The primes are listed in Appendix A. A precursor of an miR92a1/2 sequence was inserted into the pLVX vector (Invitrogen, Carlsbad, CA, USA). Lentivirus packaging was performed for HEK-293T cells by co-transfecting them with a lentiviral packaging plasmid (psPAX2, pMD2.G, Invitrogen), and expression vectors containing target genes or an empty pLVX vector (Invitrogen) used Lipofectamine 2000 Transfection Reagent (Thermo Fisher Scientific, San Jose, CA, USA) according to the manufacturer’s protocol. An empty vector was used as the negative control. SKM1 was infected using the lentivirus supernatant and selected by 2 μg/mL puromycine (Sigma-Aldrich, St. Louis, MO, USA).

### 4.7. Quantitative Realtime PCR (qRT-PCR)

Total RNA was isolated using an RNA pure Tissue & Cell Kit (CW Biotech, Beijing, China), and cDNA was synthesized using a ReverTra Ace qRT-PCR RT Kit (Toyobo, Osaka, Japan) and miRNA cDNA Synthesis Kit, with Poly(A) (Applied Biological Materials, Richmond, BC, Canada) as per the manufacturer’s protocol. Amplification and detection were performed with Power SYBR Green Master Mix (Cwbiotech) and Gentier 48R System (Tianlong Technology, Xi’an, China). The primers are listed in Appendix A. For quantitative analysis, the relative mRNA levels of β-catenin, SMAD4, ZEB2, KLF4 and PTEN were normalized to GAPDH, and the relative miRNA level of miR92a was normalized to U6. The primers are listed in Appendix A.

### 4.8. Western Blotting

Cells were collected and lysed with lysis buffer (10 μL PMSF in 1 mL RIPA). Protein concentration was determined using a BCA Protein Assay Kit (Beyotime). For Western blotting, proteins (30 μg for total cell proteins; 10 μg for exosomal proteins) were separated by electrophoresis in 10% polyacrylamide gel and transferred onto PVDF membranes. Membranes were blocked with 3% bovine serum albumin (BSA, Sigma-Aldrich, St. Louis, MO, USA), and incubated with antibody against Alix, Calnexin (Santa Cruz Biotech, Santa Cruz, CA, USA), and TSG101, CD63, CD81 (Abcam, Cambridge, UK), β-catenin, tubulin or PTEN (Cell Signaling Technology, Beverly, MA, USA), followed by the addition of HRP-conjugated secondary antibody conjugated with horseradish peroxidase (HRP, Beyotime). Bands were visualized with a chemiluminescence kit and photographed using a bioluminescence imaging system (Tanon 4600, Shanghai, China).

### 4.9. Flow Cytometric Analysis

Apoptosis and proliferation were analyzed by flow cytometry (ACEA Biosciences; San Diego, CA, USA). MDS clone cells were treated with 10 μg Exos and 2 μM Ara-C for 48 h. For apoptosis analysis, cells were stained with an Annexin V-PE kit (BD Biosciences, CA, USA). For proliferation analysis, cells were stained with an EdU Alexa Fluor 647 kit (Keygen, Nanjing, China), as per the manufacturer’s protocol.

### 4.10. Luciferase Activity Assay

The miR-92a-3p binding sites were predicted using an online website (http://mirdb.org/ accessed on 5 March 2022). The miR-92a-targeted wild-type PTEN 3′UTR and mutated PTEN 3′UTR were amplified via PCR and inserted into luciferase reporter plasmids psi-Check2, termed psi-Check2-PTEN-WT and psi-Check2-PTEN-mut. Primers are shown in Appendix A. Next, psi-Check2-PTEN-WT or psi-Check2-PTEN-Mut and miR-92a mimic or the negative control miR-NC were co-transfected into HEK-293T cells. After transfection for 48 h, luciferase activity was determined using the Dual Luciferase Reporter Gene Assay Kit (Beyotime).

### 4.11. In Vivo Mice Experiment

Female B-NSG mice (NOD-Prkdc scid IL2rg tm1/Bcgen, NSG) were irradiated with 180 cGy. A total 2 × 10^6^ SKM1 cells were injected into NSG mice through the tail vein. Mice were divided into 3 groups randomly and were injected intravenously with 1.5 mg/kg Exos derived from SKM1, SKM1-miR92a1 and SKM1-miR92a2 cells three times a week. After injection with Exos for 14 days, NSG mice were injected intraperitoneally with Ara-C (80 mg/kg) three times a week. The SKM1 in peripheral blood were stained by antibody against hCD45 and analyzed by flow cytometry after injecting Ara-C 6 times. Mice were humanely sacrificed, and spleen and femur bone were collected and stored for further experiments.

### 4.12. Statistical Analysis

All data were statistically analyzed using GraphPad Prism 5.0 (GraphPad software, San Diego, CA, USA). Data are presented as mean ± SEM. The statistical significance of differences between the means of two groups was evaluated by Student’s *t*-test. Multiple group comparisons were evaluated by ANOVA with Bonferroni’s post hoc test.

## Figures and Tables

**Figure 1 biomolecules-12-01448-f001:**
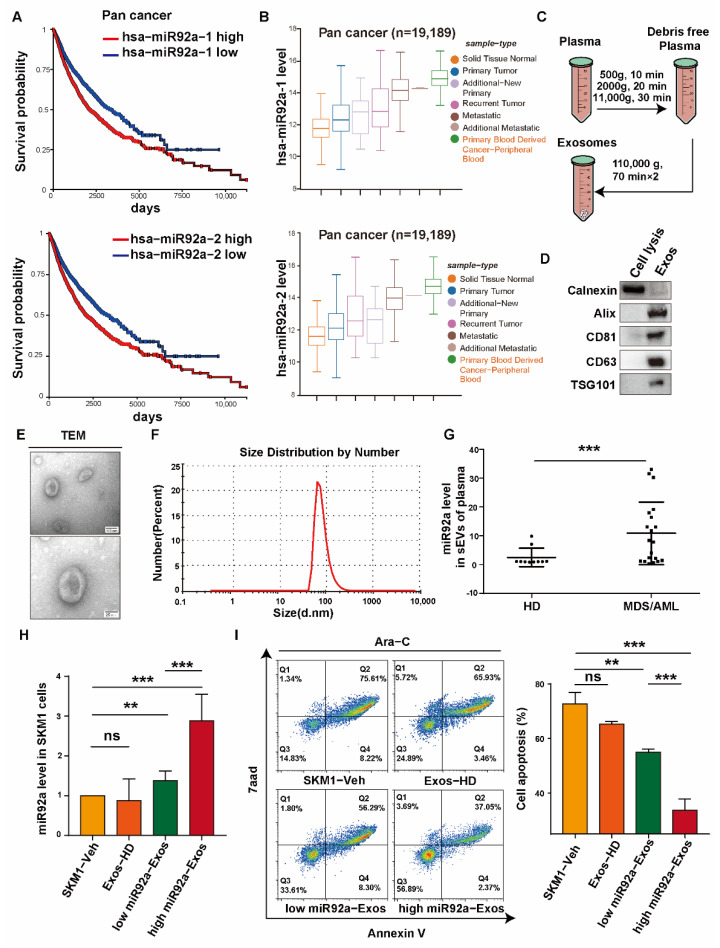
Exosomal miR92a is upregulated in MDS/AML plasma. (**A**) Kaplan–Meier survival analysis of patients with pan cancer was conducted according to the expression of miR92a1/2 using the log-rank test via the TCGA database. (**B**) miR92a1/2 level in different types of cancer analysis by TCGA database. (**C**) The scheme of exosome isolation from plasma. (**D**) Detection of Exo markers by Western blotting. (**E**) Examination of Exo morphology by TEM. (**F**) Particle size of Exos by NTA. (**G**) Expression of exosomal miR92a in plasma derived from healthy donors (HD, *n* = 10) and MDS/AML patients (MDS/AML, *n* = 20). (**H**) The ratio of miR92a level in patient plasma to miR92a level in healthy donor plasma ≥ 2 was considered as high Exos-miR92a, and <2 was considered as low Exos-miR92a. miR92a level in SKM1 treated by Exos-HD, Exos-miR92a high and Exos-miR92a low. (**I**) Apoptosis of SKM1 treated by 2 μM Ara-C and 10 μg Exos-HD, high Exos-miR92a and low Exos-miR92a for 48 h analyzed by flow cytometry. Ns ≥ 0.05, ** *p* < 0.01, *** *p* < 0.001.

**Figure 2 biomolecules-12-01448-f002:**
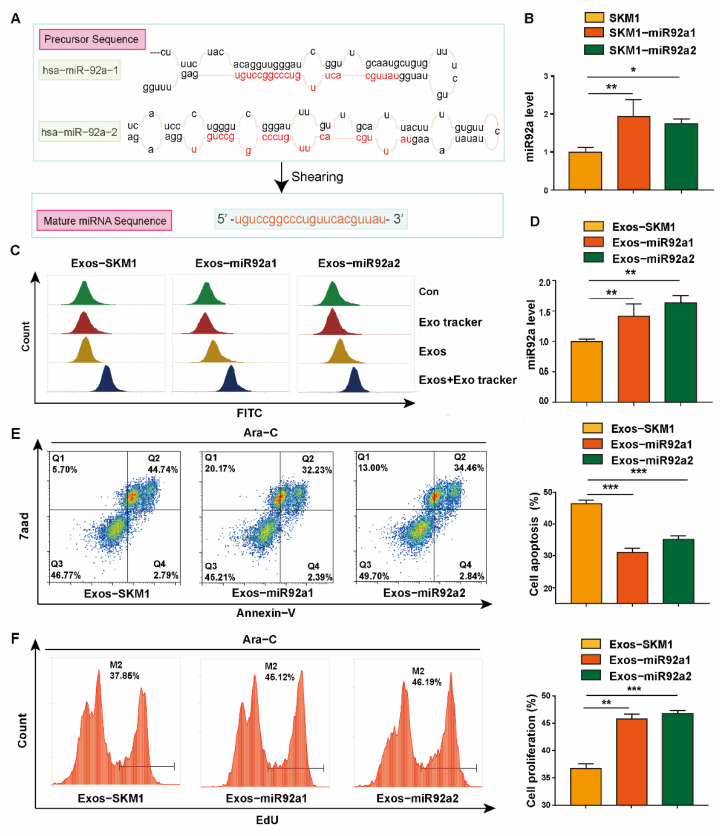
Exosomal miR92a increased resistance to Ara-C of recipient cells. (**A**) Two precursors of miR92a. (**B**) miR92a level in SKM1 and SKM1-miR92a1/2 cells. (**C**) Uptake of Exos-SKM1 and Exos-miR92a1/2 in SKM1 analyzed by flow cytometry. (**D**) miR92a level in SKM1 treated with Exos-SKM1, Exos-miR92a1/2 for 48 h. (**E**,**F**) Apoptosis (**E**) and proliferation (**F**) of SKM1 treated by 2 μM Ara-C and 10 μg Exos-SKM1 or Exos-miR92a1/2 for 48 h analyzed by flow cytometry. * *p* < 0.05, ** *p* < 0.01, *** *p* < 0.001.

**Figure 3 biomolecules-12-01448-f003:**
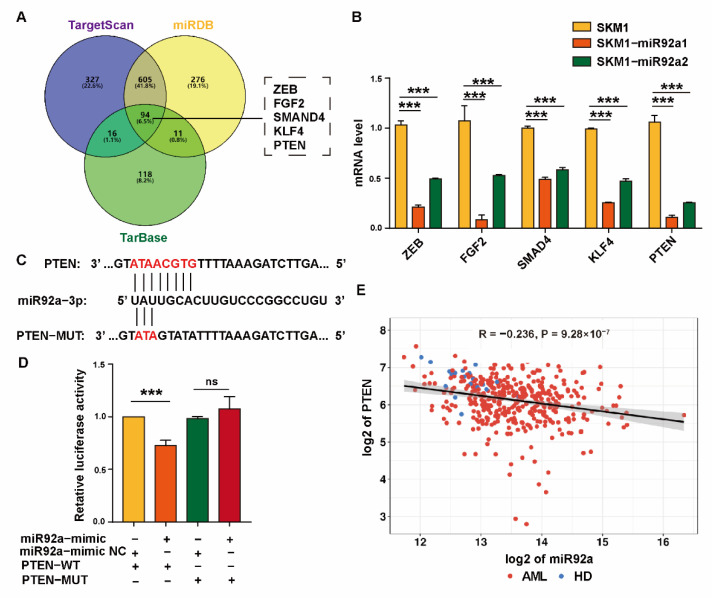
miR92a modulates PTEN expression. (**A**) Venn diagram of miR92a target genes predicted by three databases (TargetScan, miRDB and TarBase). (**B**) mRNA level of potential target genes in SKM1 treated by Exos-SKM1 and Exos-miR92a1/2. (**C**) The putative miR92a binding sites on WT sequence and mutant sequence of PTEN 3′-UTR. (**D**) HEK-293T cells were cotransfected with miR92a mimic or mimic-NC and two reporter plasmids psiCHECK2 (WT or mutant PTEN 3′-UTR sequence), and luciferase activities of transfected cells were assayed. (**E**) Correlation of expression of miR92a and PTEN in AML (*n* = 207) and healthy donors (*n* = 18), ns ≥ 0.05, *** *p* < 0.001.

**Figure 4 biomolecules-12-01448-f004:**
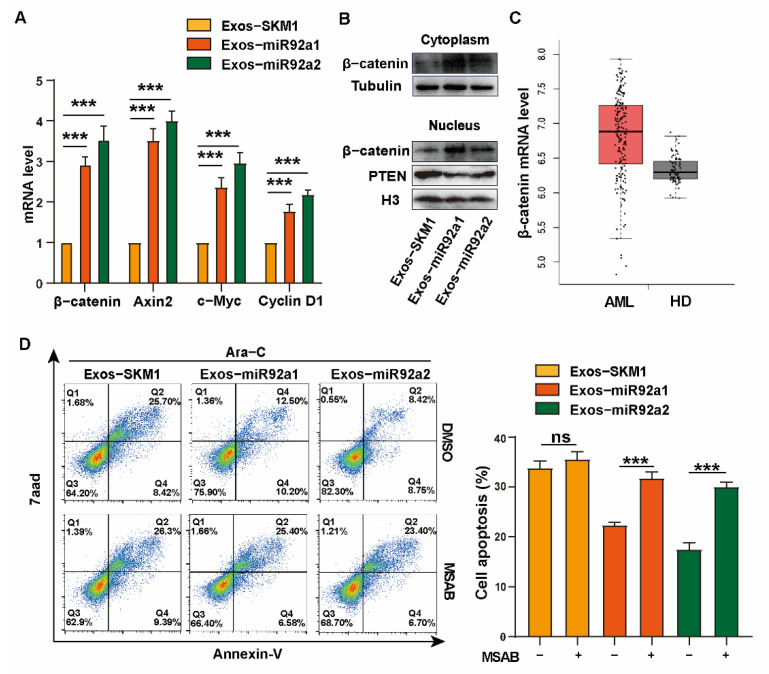
miR92a regulates cell growth via activating Wnt/β-catenin signal. (**A**) Expression of β-catenin, Axin2, c-Myc and Cyclin D1 in Exos-SKM1-, Exos-miR92a1-, and Exos-miR92a2-treated SKM1 cells analyzed by qRT-PCR. (**B**) β-catenin level in cytoplasm and PTEN and β-catenin in nucleus in Exos-SKM1, Exos-miR92a1, and Exos-miR92a2-treated SKM1 cells analyzed by Western Blotting. (**C**) TCGA analysis of β-catenin level in AML. (**D**) Apoptosis of Exos-SKM1-, Exos-miR92a1-, and Exos-miR92a2-treated SKM1 under 2.5 μM MSAB and 2 μM Ara-C treatment for 48 h, respectively, ns ≥ 0.05, *** *p* < 0.001.

**Figure 5 biomolecules-12-01448-f005:**
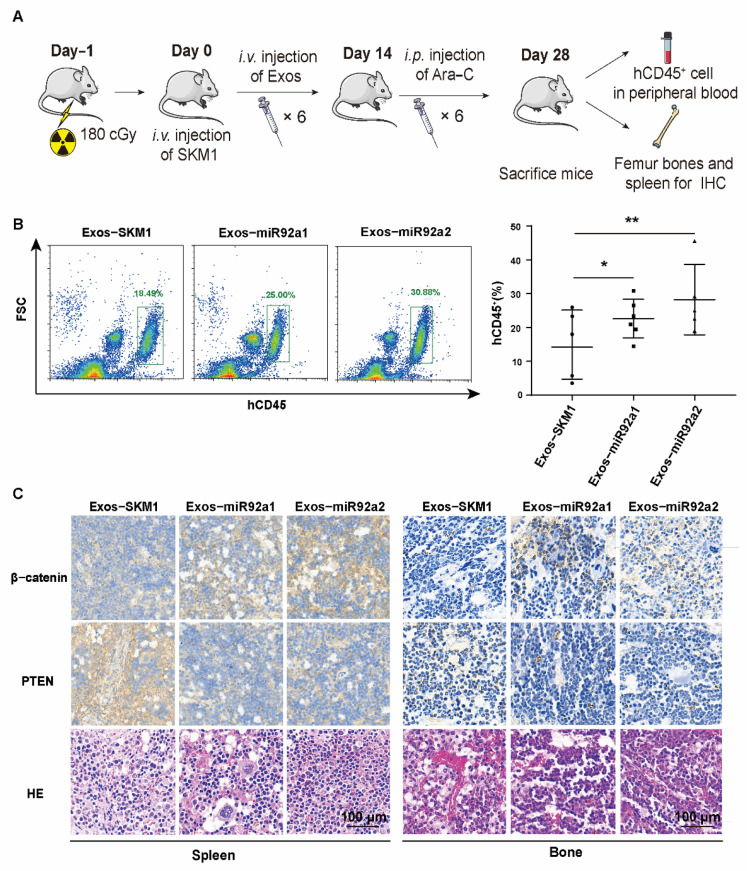
Exosomal miR92a promoted MDS/AML resistance in Ara-C in vivo. (**A**) Schematic diagram of the process for establishing mouse model. Adult NSG mice were irradiated with 180 cGy and injected with 2 × 10^6^ SKM1 cells by intravenous injection. After SKM1 injection, 1.5 mg/kg Exos-SKM1, Exos-miR92a1 or Exos-miR92a1 was injected intravenously 6 times. An amount of 80 mg/kg Ara-C was injected intraperitoneally three times a week after SKM1 injection for 14 days. The peripheral blood was analyzed by flow cytometry on day 28. (**B**) Percentages of SKM1 (hCD45^+^) in peripheral blood of mice injected with Exos-SKM1 or Exos-miRNA92a1/2 and Ara-C. (**C**) β-catenin and PTEN level in the spleen and bone tissues detected by IHC staining. Scala bar = 100 μm, * *p* < 0.05, ** *p* < 0.01.

## Data Availability

The data presented in this study are available on request from the corresponding author.

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
