# Peer review of "Exosomal miR92a Promotes Cytarabine Resistance in Myelodysplastic Syndromes by Activating Wnt/β-catenin Signal Pathway"

_biomolecules, 2022, doi:10.3390/biom12101448_

Round 1

Reviewer 1 Report

The authors demonstrate the role of exosome miRNAs, a small non-coding mRNA, particularly exosomal miR92a in MDS/AML cells. The paper is multi-step laboratory study in which authors focus on miR92a as a potential MDS and AML biomarker. Li and others showed higher expression of exosomal miR92a in MDS/AML plasma compered to healthy donors. Treating recipients’ cells with exosomes derived from MDS/AML plasma, the authors found exosomes that expressed high miR92a induced stronger proliferation ability and cytarabine tolerance. 

In my opinion the result of study presented in the paper is of importance in the field of hematological cancers development, as well as development of drug-resistance in cancers cells during oncological therapies. 

What was the final dose of mouse radiotion? 300 cGy or 180 cGy? There is incompatibility in Figure 5 between the text and the picture. Later in the text (p. 4.11) dose 180 cGy is mentioned. 

There are some typos mistakes in the text. Unification of abbreviation “Ara-C  / Arc-a”  is need. The mild English editing is needed. Editing of the text is also needed – spaces, different font size, etc. 

Author Response

Reviewer #1:

The authors demonstrate the role of exosome miRNAs, a small non-coding mRNA, particularly exosomal miR92a in MDS/AML cells. The paper is multi-step laboratory study in which authors focus on miR92a as a potential MDS and AML biomarker. Li and others showed higher expression of exosomal miR92a in MDS/AML plasma compered to healthy donors. Treating recipients’ cells with exosomes derived from MDS/AML plasma, the authors found exosomes that expressed high miR92a induced stronger proliferation ability and cytarabine tolerance.

In my opinion the result of study presented in the paper is of importance in the field of hematological cancers development, as well as development of drug-resistance in cancers cells during oncological therapies.

Comment 1: What was the final dose of mouse radiotion? 300 cGy or 180 cGy? There is incompatibility in Figure 5 between the text and the picture. Later in the text (p. 4.11) dose 180 cGy is maentioned.

Response: We apologize for incorrect schematic diagram in regard to the radiation dose. (Figure 5A). 180 cGy is the dose for radiation. Please find the revised version of Figure 5A.

Comment 2: There are some typos mistakes in the text. Unification of abbreviation “Ara-C / Arc-a” is need. The mild English editing is needed. Editing of the text is also needed – spaces, different font size, etc.

Response: The manuscript was greatly revised, and the typos have been revised.

Reviewer 2 Report

In this manuscript, the authors investigated the effects of exosomal miR92a on Ara-C resistance in myelodysplastic syndrome (MDS). I would like the authors to address the following issues.

1.      Figure 1H and I: Statistical comparison between low and high miR92a exosomes should be performed.

2.      Figure 1I: Patients-derived exosomes containing low miR92a can induce Ara-C resistance, but to a relatively lower extent than high miR92a exosomes. This point should be briefly mentioned and discussed regarding potential exosomal factors other than miR92a.

3.      Please revise the legend of Supplementary Figure 2CDE. 

4.      ref. 26 should be removed as it is a retracted paper.

5.      Figure 4A: It is doubtful that there exists a significant difference between Exos-SKM1 and Exos-miR92a1. In addition, as transcriptional targets of β-catenin, AXIN2, DKK1, cyclin D1, MYC etc. should be examined.

6.      It is somehow difficult to explain how IWR-1 blocks the effect of miR92a. IWR-1 inhibits Axin protein destruction while PTEN activates GSK3β by suppressing AKT. Axin and GSK3β, in additino to APC, form a destruction complex to induce proteasome degradation of β-catenin. Thus, even though IWR-1 activates Axin, the destruction complex is no longer formed because miR92a suppresses GSK3β. This point should be solved by rationalizing it from previous research papers, or by using a direct β-catenin inhibitor or knockdown.

7.      Figure 5C: Images of β-catenin and PTEN are not clear enough to see the difference. Please replace them with ones with higher resolution.

8.      Figure 5A: Please check radiation dose as it is 180 cGy in the legend and Materials and Methods section.

9.      Please discuss the donor cell of exosomes containing miR92a in MDS/AML.

Author Response

Responses to comments of reviewer

Reviewer #2:

In this manuscript, the authors investigated the effects of exosomal miR92a on Ara-C resistance in myelodysplastic syndrome (MDS). I would like the authors to address the following issues.

Comment 1: Figure 1H and I: Statistical comparison between low and high miR92a exosomes should be performed.

Response: Statistical comparison between low and high of miR92a in plasma exosomes in MDS has been added in Supplementary Fig. 1A.

Comment 2: Figure 1I: Patients-derived exosomes containing low miR92a can induce Ara-C resistance, but to a relatively lower extent than high miR92a exosomes. This point should be briefly mentioned and discussed regarding potential exosomal factors other than miR92a.

Response: Thanks for your comments. We have briefly discussed this point in Discussion section of revised MS.

Comment 3: Please revise the legend of Supplementary Figure 2CDE.

Response: We apologize for incorrectly writing. The legends of Supplementary Figure 2C-E has been revised.

Comment 4: Ref. 26 should be removed as it is a retracted paper.

Response: Ref. 26 has been removed.

Comment 5: Figure 4A: It is doubtful that there exists a significant difference between Exos-SKM1 and Exos-miR92a1. In addition, as transcriptional targets of β-catenin, AXIN2, DKK1, cyclin D1, MYC etc. should be examined.

Response: Thanks for your suggestions. mRNA levels of β-catenin and Wnt/β-catenin pathway target genes (Axin2, Cyclin D1 and c-Myc) in Exo-SKM1 and Exos-miR92a1/2 treated SKM1 cells and ML1 cells were analyzed by real-time PCR. The results were shown as in Figure 4A and Supplementary Figure 3A of revised MS.

Comment 6: It is somehow difficult to explain how IWR-1 blocks the effect of miR92a. IWR-1 inhibits Axin protein destruction while PTEN activates GSK3β by suppressing AKT. Axin and GSK3β, in additino to APC, form a destruction complex to induce proteasome degradation of β-catenin. Thus, even though IWR-1 activates Axin, the destruction complex is no longer formed because miR92a suppresses GSK3β. This point should be solved by rationalizing it from previous research papers, or by using a direct β-catenin inhibitor or knockdown.

Response: Thanks for this helpful suggestion. We choose 2.5 μM MSAB (HY-120697) (MedChemExpress, New Jersey, USA), a selective inhibitor of Wnt/β-catenin signaling, which binds to β-catenin, promoting its degradation to downregulated β-catenin level to treat recipient SKM1 cells. And the apoptosis was analyzed. We found under MSAB treatment, β-catenin levels in Exos-SKM1 and Exos-miR92a1/2 treated SKM1 and ML1 were decreased (See the Figures in following panel A and B). And when SKM1 and ML1 were incubated with MSAB, apoptosis of Exos-miR92a1/2 treated recipient cells was increased significantly compared to Exos-SKM1 (Fig 4D & Fig S3D).

Comment 7: Figure 5C: Images of β-catenin and PTEN are not clear enough to see the difference. Please replace them with ones with higher resolution.

Response: The higher resolution images of β-catenin and PTEN has been replaced in the revised MS (Figure 5C).

Comment 8: Figure 5A: Please check radiation dose as it is 180 cGy in the legend and Materials and Methods section.

Response: We apologize for incorrect schematic diagram in regard to the radiation dose (Figure 5A). 180 cGy is the dose for radiation. Please find the revised version of Figure 5A.

Comment 9: Please discuss the donor cell of exosomes containing miR92a in MDS/AML.

Response: Thanks for your suggestions. We discussed this point in the update

Round 2

Reviewer 2 Report

I appreciate the authors to adequately address my comments. However, in my comment #1 in the previous review round, I had suggested to statistically compare the miR92a levels and apoptotic cell fractions between SKM1 cells treated with low and high miR92a-Exos. This point must be clarified to underline the pathological significance of exosomal miR92a and would be of interest for most readers.

Author Response

Responses to comments of reviewer

Reviewer #2:

 Comment 1: I appreciate the authors to adequately address my comments. However, in my comment #1 in the previous review round, I had suggested to statistically compare the miR92a levels and apoptotic cell fractions between SKM1 cells treated with low and high miR92a-Exos. This point must be clarified to underline the pathological significance of exosomal miR92a and would be of interest for most readers.

Response: Thanks for your suggestions. Statistical comparison of the miR92a levels and apoptotic rates between SKM1 cells treated with low and high miR92a-Exos has been added in Fig. 1H & I.
